# Business Models 4.0 Using Network Effects: Case Study of the Cyfrowy Polsat Group

**Jerzy Niemczyk, Rafał Trzaska, Maciej Wilczyński and Kamil Borowski ***

Department of Strategy and Management Methods, Wroclaw University of Economics & Business, 118/120 Komandorska Street, 53-345 Wroclaw, Poland; jerzy.niemczyk@ue.wroc.pl (J.N.); rafal.trzaska@ue.wroc.pl (R.T.); maciej.wilczynski@ue.wroc.pl (M.W.)
* Correspondence: kamil.borowski@ue.wroc.pl

**Abstract:** The aim of the research, the results of which are presented in the article, is to discover new knowledge allowing for the description and design of business models 4.0 using network effects. In their research, the authors reviewed the literature, carried out desk research and conducted an analysis of publicly available documents of selected companies from the Cyfrowy Polsat Group. The results of the study present the possibility of using the network effect in business models 4.0. The paper develops a framework for business model analysis from the perspective of Industry 4.0. The presented research will allow for an indication of the possibility of using a business model from the perspective of Industry 4.0, based on the theory of the network effect in building the value of network organizations.

**Keywords:** business model; Industry 4.0; network strategy; strategic management

## 1. Introduction

The authors of this publication have been conducting research in the areas of strategies and business models for many years. Their experiences, and a critical review of the literature on the subject, show that the lack of business models dedicated to companies operating in Industry 4.0 is a significant research subject. Business models in new technology sectors are an attractive research object for management theorists and practitioners. This can be seen in the number of scientific papers on business models and in the number of studies related to the impact of Industry 4.0 on management, as well as in the popularity of this issue in business and industry magazines. While so-called business models 4.0, in terms of technology, are constantly enriched with new solutions, in the field of social sciences—including management sciences—business models 4.0 require continuous research [1].

The aim of the article is to discover new knowledge which allows for the description and design of business models from the perspective of Industry 4.0. In particular, the goal is to identify business models 4.0 that use the theory of network effects that are specific to network organizations. It is worth emphasizing that the network effect is integrally related to the basic goals of the organization's development, referring to the search for synergy from combining resources. In the case of the network effect, the benefit of using a good/service increases with the number of users adopting the same or compatible good/service [2]. It is worth emphasizing that the use of the network effect fits into the scope of Industry 4.0 in a special way. Without technology 4.0, it is not possible to achieve all of the advantages of network effects.

The results presented in the article try to indicate the possibility of using a business model from in the perspective of Industry 4.0, based on the theory of the network effect in building the value of network organizations. The analysis will use a case study to illustrate the logic of operation of this type of model in an organization that has been declaring network activities for many years.

The authors were trying to answer the following research questions:

1. Which of the 4.0 technologies build the value of the organization in the business model and how?
2. How does the business model 4.0 increase the possibilities of achieving network effects?
3. How have business models in the communication services sector evolved?
4. How does the media group use business model 4.0 to build its value?
   a. Are there events in the study group from the perspective of events enabling the identification of BM 4.0 operating on the basis of a network effect?
   b. Is it possible to identify the size of the network effect in BM 4.0 in the studied group?
   c. Is there any relationship between changes in the group's business model (from the perspective of the Cyfrowy Polsat Group) and selected stock market indicators?

Empirical research was conducted based on the case study methodology for the Cyfrowy Polsat Group (The subject of the case study is Cyfrowy Polsat S.A.—company listed on the Warsaw Stock Exchange (WSE: CPS)). The selection of this group resulted from the assumption that the capital group operating in the media market and other new technology markets is subject to pressure from technology 4.0 and it already uses network effects in its activities. The Cyfrowy Polast Group also operates in the energy, banking, insurance and real estate sectors. Therefore, the proposed case study of the Cyfrowy Polsat Group will also serve as a study on the possibility of using the network effect in class 4.0 in the description and understanding of the business model. The study will cover the following areas: the phase of diagnosis of the surveyed organization from the perspective of events enabling the identification of the business model under study, the phase of analysis, assuming the study of the network effect in the Cyfrowy Polsat Group's business model, and the phase of assessing and verifying the conclusions obtained through the use of stock exchange indicators. The choice of the Cyfrowy Polsat Group was also dictated by the fact that it is great example of the BM 4.0 model, using the network effect in organizations from the sectors, other than just media services.

In their research, the authors reviewed the literature, carried out desk research, and conducted an analysis of publicly available documents on selected companies of the Cyfrowy Polsat Group.

## 2. Business Models 4.0

### 2.1. Business Model 4.0 and Value Building

A Business Model (BM) is a concept that has been the subject of considerable research in management and quality sciences. This research was initiated more widely in the period of domination in the management of the Resource-Based View theory. This was in the 1990s and in the first decade of the 21st century [3]. It was then that the following definition of a business model was created, according to A. Osterwalder and Y. Pigneur: "A business model describes the rationale of how an organization creates, delivers, and captures value" [4]. With this definition, the above-mentioned authors named increasing value as the goal of building a business model. The original value is described from the customer's perspective. Admittedly, in their work "An eBusiness model ontology for modelling eBusiness", A. Osterwalder and Y. Pigneur tried to consider newer trends in business, pointing to: "(1) The products and services a firm offers, representing a substantial value to the customer, and for which he is willing to pay. (2) The infrastructure and the network of partners that are necessary in order to create value and to maintain a good customer relationship. (3) The relationship capital the firm creates and maintains with the customer, in order to satisfy him and to generate sustainable revenues. And last, but not least, (4) the financial aspects" [5]; however, their definition of the business model has remained unchanged and is still the most frequently cited.

The business model in its colloquial, entrepreneurial form is a description of the way a company earns money, while in science it is usually the same as the means of building value [6,7].

A. Jabłoński and M. Jabłoński emphasized several types of business model decomposition and presented almost 40 different definitions of the business models identified in the literature for the period of 1998–2017, together with the presentation of the main term in relation to configuration management [8].

The concept of BM has evolved. In the first phase, the value to which BM was sought was equivalent to gross profit. In turn, in the period of the domination of the Value-Based Management approach, it was already an economic value, understood as an increase in the wealth of business owners measured by the increase in the value of shares/stocks. Now, thanks to the model popularized by A. Osterwalder and Y. Pinquer [4], value is broadly understood as the utility generated in the company for its stakeholders.

Numerous ways of generating value were identified in economics and management, and the concept of economic rent is frequently used in the description of these methods. The most important economic rents indicated in the context of BM include:

- Ricardian rent, i.e., a rent for the right to obtain income from the use of own and scarce resources;
- Rent due to the economies of scale and scope that allows for an increase in the economy of production by increasing the scale of production;
- Chamberlin rent, i.e., rent indicating income from taking a monopolistic position;
- Schumpeterian rent, i.e., rent that provides the right to benefit from innovation, especially from disruptive innovation [9,10].

Aside from the aforementioned types of rent, there are also others [11]. Economic rents used or implied by the Industry 4.0 concept may be of particular interest (Figure 1). Industry 4.0 allows one to achieve, above all, rent due to the economy of scale, with the difference that, unlike classical production processes, this rent in Industry 4.0 can be achieved thanks to a much lower profitability threshold (compare with the Blitzscaling theory [12]). Industry 4.0 is also a provider of network effect, long tail effect and Big Data effect (this is part of the concept widely described in the article "Scalability 4.0 as economic rent in Industry 4.0" [13]). The model, presented in Figure 1, shows the relationship between selected Industry 4.0 technologies, elements of the business model, as discussed by A. Osterwalder and Y. Pigneur (2010), and the methods of value delivery (Value Proposition, Value Creation, Value Delivery, Value Capture, Value Communication, and Value Proposition), as well as economic rents, which make up the Scalability 4.0 concept.

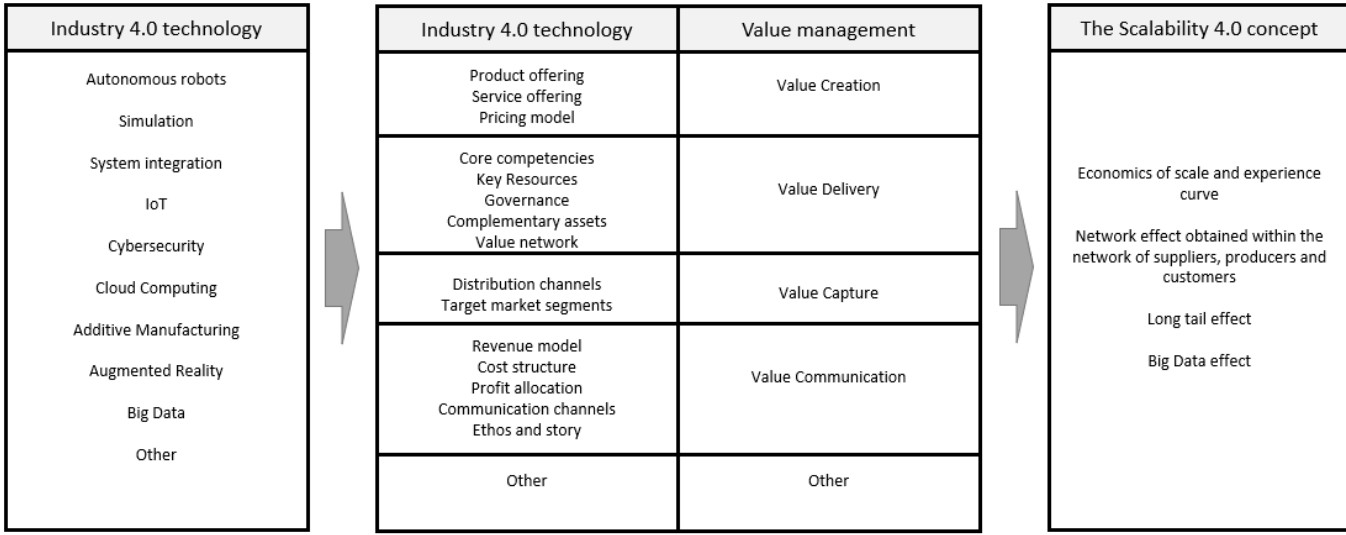

**Figure 1.** The concept of combining technology, BM elements and value proposition; source: own.

In addition to the methods of building the values indicated in Figure 1, in order to more precisely indicate the features of the business model 4.0, based on Industry 4.0, it is

worth referring to other business models and characteristics that were created only because the 4.0 dimension appeared in economy and management.

The analysis of the Industry 4.0 business model should begin with the definition of revolution 4.0. The research assumed that Industry 4.0 is a set of techniques and technologies identical to the digital transformation of the economy. The most common technologies mentioned here are cybersecurity, simulation, augmented reality, system integration, Internet of Things, additive manufacturing, autonomous robots, Big Data, artificial intelligence, and cloud computing. The above-mentioned technologies are strictly engineering categories. Some authors deepen the concept of Industry 4.0, introducing other tools, issues and concepts that are part of the general domain of the subject, trying firstly to improve our understanding of it and secondly to link it with topics of interest on managerial, engineering and information technology level; predominantly, in relation to resource efficiency, shortening innovation cycles and raising productivity [14]. The literature on the subject includes research showing that Industry 4.0 also affects changes at the levels of economy, management systems, as well as cultural and social changes [15] and others, such as low-cost automation, robotics, 3D printing, smart factory or block chain [16]. Moreover, it is a necessary tool when it comes to maintaining a competitive advantage [17,18]. The abovementioned new technologies introduced by Industry 4.0 have a significant impact on the creation of new and adaptation of existing business models, e.g., Internet of Things [19], Industrial Internet of Things [20], Cyber–Physical Systems (CPS) [21,22], Big Data [23]. M. Dobrowolska and L. Knop claim, that "Business model changes are the expression of companies' hunger for gaining competitive edge and the effectiveness of operations". The Industry 4.0 concept has strengthened changes in business models, especially in relation to the creation of value based on new technologies, options for creating unique resources, key competencies and treating business models as networks of mutually dependent activities and processes [24].

The literature overview conducted by D. Ibarra, J. Ganzarain and J. Igartua-Lopez [25] presents the main design principles of the Industry 4.0, challenges to traditional business models and key challenges to digital transformation in global business. The design principles allow companies to investigate a potential transformation towards Industry 4.0 technologies. One of these principles is "interoperability", which means that machines and people need to be able to communicate through the Internet of Things or "Real-Time Capability" when smart factories need to be able to capture, store and analyze data in real time to make immediate decisions according to new findings. Among the other principles that the authors have included, there are decentralization, virtualization, service-orientation and modularity [26]. The main challenges related to traditional BM refer to networking, customized mass production, low price, local production, fragmentation of the value chain or the decentralization of production. The authors summarized the main prerequisites to digital transformation: appropriate work organization, availability of products, new business models, talented workers, investment in R&D or legal framework.

The Industry 4.0 concept offers opportunities to create new, or redesign, existing business models based on new products and services and ways to serve clients, fulfill their needs, as well as improving efficiency and integration across the entire value chain. Tirabeni emphasizes three main research streams that emerged in their analysis: customer and service centered business models, integrated and networked business models and sustainable business models, naming them as Innovative Business Models [27].

The most important effects of implementing technology 4.0 in business, based on the previously mentioned economic rent, include:

- Reduction in transaction costs for signed contracts; thanks to this, it is possible to widely implement contractual solutions instead of hierarchical solutions;
- Achieving economies of scale on conditions that are previously unattainable, thanks to the longtail rule, the possibilities of Big Data and Network Effect [13];
- Simplification of business processes by resigning from intermediaries thanks to 4.0 technologies;

- Obtaining the effects of the continuous generation of innovation through the use of an open innovation system [28];
- Effects of the "the real-time processing and use of big data for consumer research and marketing, as well as for security purposes, the rapidly growing importance of millennials as a customer group, the sharing economy, and direct digital customer contact from companies without intermediaries" [29];
- Obtaining the effects that are specific to the complexity theory: self-organization, dependency paths, complex adaptive systems, and the selection context specific to the complexity theory [30].

From the Industry 4.0 perspective, the business model must be perceived differently. In the classic canvas model, there was assumed a structural and functional arrangement of nine business model elements. In the cause-and-effect system, these elements were to determine the value proposition, but above all, it was the value proposition that was to influence the shape of these nine elements. In the business model 4.0, the subject of the analysis is: the network of various connection methods, the effect of servicization, methods of generating various value streams (standardization in diversity), methods of generating value based on a uniform communication protocol without transaction costs [31]. It is, therefore, worth considering such changes in the definition of the business model so that they better reflect the impact of Industry 4.0.

### 2.2. Business Model 4.0 Using Network Effects

Business model 4.0 was defined in the previous part of this article. It has features resulting from the openness of management to 4.0 technologies. To be able to define BM 4.0 additionally using network effects, it is necessary to first adopt a specific definition of this effect.

The network effect has its origins in the development of telephone services at the beginning of the 20th century. It was then popularized by the invention of Ethernet and, for many years, was associated with telecommunication, information and IT infrastructure. The economic theory of the network effect was advanced significantly from 1985 [2]. There is not much research conducted on the network effect as the catalyst for a change in the business models and as a tool to build the competitive advantage. The authors explored this effect in their previous papers [13,32,33].

Network effect is defined as the phenomenon that the benefit of using a good/service increases with the number of users adopting the same or compatible good/service [2]. Ch. P. Lin and A. Bhattacherjee (2008) cite M. L. Katz, C. Shapiro, J.M. Gallaugher and Y.M. Wang "that network effects arise when the utility that consumers derive from the consumption of a product or service depends on the number of other users of the same product or service or the availability of complementary products or services that generate additional value for users of the original product or service" [2,34]. Factors that drive network effects, such as network size and availability of complementary goods or services, are called network externalities [35], and products or services exhibiting such effects are called network goods [36]. There are two types of network externalities: direct and indirect. "Direct network externalities are based on the number of participants in a given network. ( . . . ) As new participants enter these networks, existing users gain more choice in terms of trading, communicating, or playing games, and thus gain network utility. In contrast, indirect network externalities are ancillary benefits accruing to network participants as a network grows, such as the development of complementary services, standards formation, and price reduction, but not directly from the number of network participants" [36] ("Since MIM (Mobile instant messaging) is essentially a communication technology, network effect should be able to explain the user-perceived values of MIM apps, thus capable of developing hypotheses herein. That is, users can perceive more value from an MIM app if more users adopt the same app, because (1) the user can communicate with more peer users (i.e., the direct network effect), and (2) more value-added features and complementary offerings will be available from the MIM provider or third-party developers (i.e., the

indirect network effect). Therefore, if users perceive more network effect from a certain MIM app, they would perceive more value from that app" [37]). Network externalities are qualities of certain goods and services that make them more valuable to a user as the number of users increases [38].

In this form, BM 4.0 using network effects must be composed in such a way that technologies 4.0 provide it with appropriate feature, specific to Industry 4.0, listed in the second column (Table 1) and ensure that these features simultaneously positively influence the creation of the direct and indirect network effects indicated in the third column (Table 1).

**Table 1.** The concept of combining technology 4.0, BM 4.0 elements, and BM 4.0 plus the network effect.

| Technology 4.0 | Features of Business Models 4.0 | Business Model 4.0 Using Network Effects |
|---|---|---|
| • Cybersecurity;<br>• Simulation;<br>• Augmented reality;<br>• System integration;<br>• Internet of Things;<br>• Additive manufacturing;<br>• Autonomous robots;<br>• Big Data;<br>• Artificial intelligence;<br>• Cloud computing; | • They must link value streams (merge transactions) by eliminating all transaction cost drivers;<br>• They must combine the existing methods of generating value on the basis of service;<br>• They must combine in such a way as to ensure the possibility of generating different streams of values and, at the same time, customizing;<br>• They must combine different methods of generating value based on a uniform communication protocol;<br>• They must generate value across the entire ecosystem;<br>• They enable the achievement of economies of scale at a lower level thanks to 4.0 technologies;<br>• They enable the achievement of the effects of continuous innovation generation through the use of an open innovation system;<br>• They enable the achievement of the effect of the real-time processing type, and the use of big data for consumer research and marketing;<br>• They enable the achievement of SaaS class effects, sharing economy, or ride sharing. | • Direct network effect occurs when a user's benefits result directly from the number of users of a particular product or service [36].<br>• Indirect network effect occurs when the benefits of using the main product come from the availability of complementary products [36]. |

Source: own.

The analysis (Table 1) shows that direct effects are possible mainly due to the elimination of transaction costs, the service of the methods of generating value, ensuring the possibility of generating various value streams and customizing, working in the ecosystem, and having a uniform communication protocol. In turn, indirect effects are achieved thanks to the economy of scale, building sharing economy platforms [39], building ride-sharing platforms [40], creating SaaS sales models, achieving real-time processing, and applying the principle of service and the classic principle of operational synergy.

The network effect can also be obtained in the course of the development strategy through mergers and acquisitions. By focusing on making further acquisitions, the company, on the one hand, is constantly growing its customer base. On the other hand, by diversifying its portfolio of services and products, it allows for taking advantage of the so-called cross-selling actions, penetrating the customer base even more. The literature review conducted by Ch. Öberg on the network literature on M&As showed that research either focuses on the M&A parties or refers to M&A effects on business partners and several papers are conceptual, arguing that "M&A effects on business relationships is an unexplored area" [41].

### 2.3. Evolution of Business Models in the Communication Services Sector

There are four main types of business models in the Communication Services sector. They are the following businesses:

- Subscription: the company sells products without advertising;
- Advertising: the product is delivered for free, but is linked with advertising;
- Mixed: subscription and advertising models operate simultaneously;

- Dual: there are two products—one with fewer ads and better content, and the other with worse content and more ads [42].

The United States can be mentioned as an example of the evolution of classic business models for the Communication Services sector companies, where the revenue streams of traditional Media and Entertainment corporations were based mainly on advertising until the end of the 20th century, additionally focusing on profit and dividends for shareholders (value in the meaning of gross profit). However, this has changed with the development of the ICT industry, especially the Internet. As vividly seen in Table 2. The model switched drastically towards digital. It was then that Internet advertisers, such as Google, AOL, and Yahoo, appeared on the Communication Services market. These companies began to offer their services for free, taking advantage of the low-cost benefits of the Internet as well as technological development, basing their revenues on mass advertisers. It was the beginning of the first revolution on this market. To this day, the Internet is the basic determinant of technology development in this area. Due to the pressure of the Internet, the traditional sectors that rely mainly on open access advertising, in particular the press, radio, and television, have been most affected by the changes. Overall, print newspapers and magazines, alongside with broadcast in radio and televisions ads market share declined by almost threefold within the last 20 years. Their share was taken by constantly growing online media presence [43].

Additional acceleration was caused by the digital revolution and the emergence of new types of Communication Services companies: companies operating as social networks (Facebook or Twitter) and companies that are social and information services (Buzzfeed), which began to take over the advertising market share. The next stage is the emergence of streaming technology and mobile devices. Here, there is also perceptible influence of the increasingly popular revolution 4.0 services, based on the subscription model (Netflix or Spotify). These companies offer services on integrated platforms combining an increasing number of functions. The next stage will most likely be companies providing customized content and advertising services using selected 4.0 technologies outside the Internet (AI, Internet of Things, Augmented Reality, Big Data).

**Table 2.** Business models in the Communication Services sector on selected examples.

| Company Name | The Company's Business Model | Sector | the Year the Company Was Founded |
|---|---|---|---|
| New York Times | Mixed | media/press | 1922 |
| NBC | Mixed | media/TV | 1926 |
| CBS | Mixed | media/TV | 1927 |
| ABC | Mixed | media/TV | 1943 |
| HBO | Subscription | media/TV | 1972 |
| AOL | Advertising | ICT/internet | 1985 |
| Google | Advertising | ICT/internet | 1988 |
| Yahoo | Advertising | ICT/internet | 1994 |
| Netflix (DVD) | Subscription | ICT/TV | 1997 |
| Metro | Advertising | media/press | 1999 |
| Pandora | Dual | ICT/radio | 2000 |
| Facebook | Advertising | ICT/internet | 2004 |
| Huffington Post | Advertising | ICT/internet | 2005 |
| YouTube | Advertising | ICT/TV | 2005 |
| Spotify | Dual | ICT/radio | 2006 |
| Buzzfeed | Advertising | ICT/internet | 2006 |
| Netflix (streaming) | Subscription | ICT/TV | 2007 |
| YouTube (Premium) | Subscription | ICT/TV | 2018 |

Source: own.

Changes in the BM of companies from the Communication Services sector resulted in pressure from traditional Communication Services companies and their search for new business models, as well as better reaching the consumer with their content or starting more dedicated advertising activities [44]. This led to the development of completely new business models referring to the proposals contained in Section 2.1. This will particularly apply to models using network effects. Therefore, the market in which the Communication Services sector companies operate must consider models referring to service, combining different value streams and value streams based on a uniform communication protocol, to generate value across the entire network of network effects, to innovation, and to other benefits of complexity theory.

## 3. Research Procedure

The research procedure proposed in this work is based on the classic case study method [45]. The case study is a research method that allows for identifying the tested object, taking into account, e.g., cause and effect events. It is a qualitative research method. Its main goal is to understand the given phenomenon—in this case, to learn about BM 4.0 using network effects. This will provide the basis for generalizing the results of the research on Cyfrowy Polsat Group to the level of general knowledge. This is unique case and company on the polish market. In these cases, where there are no other cases available for replication, the researcher can adopt the single-case design [46,47].

The choice of Cyfrowy Polsat Group for the case study analysis was dictated by the assumption that the group operating on the Communication Services market, which is heavily explored by technology and technology 4.0, probably leverages the network effect in its value-building strategy. Cyfrowy Polsat is a group that is at the forefront of many new business solutions in the Communication Services sector and is a leader in terms of innovation and speed of adapting to new operating conditions. The group is also making strides in the ICT sector. Due to the frequent mergers and acquisitions, scale of operations, and business innovations based on the network effect, the group was selected to help discover ways to describe and design business models in the Industry 4.0 perspective.

The study of this capital group, and more precisely the study of BM 4.0 used by this group, will enable the expansion of knowledge in the field of business models based on the network effect characteristic of companies operating in Industry 4.0. The study will include:

- Diagnosis of the studied group from the perspective of events which enable the identification of BM 4.0 operating on the basis of a network effect (mergers, acquisitions, significant decisions regarding internal development); use of desk research and white interviews;
- Analysis assuming the identification and analysis of the size of the network effect in BM 4.0 in the study group; use of desk research;
- Assessment and verification of the obtained information using an analytical technique based on the study between changes in the group's business model (from the perspective of the Cyfrowy Polsat Group) and selected stock market indicators;

The survey was conducted in the first half of 2020. The following literature databases were used in the study: Scopus, Science Direct, JSTOR and Google Scholar.

## 4. Results of Own Research of a 4.0 Class Business Model Using Network Effects—Cyfrowy Polsat Group Case Study

Cyfrowy Polsat Group is currently the biggest media/ICT conglomerate in Poland. Established in 1992 as a Polish commercial TV station, it developed between 1999–2003 to create Cyfrowy Polsat—a satellite TV platform. Cyfrowy Polsat offers broadcast satellite subscription television services in Eastern and Central Europe. The company offers a variety of television and radio channels and distributes signal decoders. Since 2013, Cyfrowy Polsat has been listed on the Polish Warsaw Stock Exchange, while the majority of ownership still belongs to its first owner—Zygmunt Solorz-Żak (Zygmunt Solorz-Żak's holding as of 02.01.2021—56.95%). The year 2009 was used as a baseline year for the

analysis due to the substantial growth of the company, and the public availability of the data. Since then, Cyfrowy Polsat Group has also been exceptionally active in the mergers and acquisitions market, growing mostly through programmatic takeovers of companies from media, ICT, and energy/sustainability sectors [48]. This was in line with its strategy of distributing the content on the widest possible scale, using the most modern devices and technologies.

An in-depth analysis of press releases, company notes and commentaries allowed us to identify key reasons for M&A activities, which prove the systematic realization of network effects, and the combination and creation of different value streams and innovations, aligned with the BM 4.0. strategic approach. Among the core business premises of Cyfrowy Polsat's M&A activities are:

- Client base expansion;
- Sales channels expansion;
- Operational and financial synergy;
- Market consolidation;
- Product portfolio diversification;
- New resources acquisition;
- Economies of scale.

Each element was outlined in the Table 3 below, alongside the type of transaction, value, shares acquired and key assets/client base captured.

**Table 3.** M&A activity of Cyfrowy Polsat S.A. between 2009–2020.

| Date of Final Agreement | Target | Transaction Value (mln USD) | Shares Bought (%) | Shares Before Transaction (%) | Business Reason | Client Base/Key Assets Captured |
|---|---|---|---|---|---|---|
| 2009 | Sferia S.A. | 14.79 | 11 | 0 | Client base expansion, product portfolio diversification (Sferia SA is an Internet provider) | 89.9 B2C and 27 thousand B2B clients |
| 2010 | mPunkt Polska S.A. | N/A | 100 | 0 | Sales channels expansion | Over 200 brick and mortar stores and locations in 150 cities |
| 2011 | Telewizja Polsat Sp. z o.o. | 1301 | 100 | 0 | Operational and financial synergy, market consolidation | TV Station, shows, brand |
| 2011 | INFO-TV-FM Sp. z o.o. | 9.24 | 61.23 | 0 | Operational and financial synergy, product portfolio diversification | Radio and TV frequency acquisition |
| 2012 | Ipla | 47.95 | 100 | 0 | Market consolidation, economics of scale, product portfolio diversification | 1.4 million active users |
| 2013 | Polskie Media Amer.com S.A. | 99 | 100 | 0 | Operational and financial synergy, market consolidation | Books and newspaper publishing group |
| 2014 | Metelem Holding Co. Ltd. | 180.91 | 83.77 | 0 | Operational and financial synergy, market consolidation, economics of scale, client base expansion, product portfolio diversification | 8 million new users |
| 2014 | Metelem Holding Co. Ltd. | 303.95 | 16.23 | 0 | Operational and financial synergy, market consolidation, economics of scale, client base expansion, product portfolio diversification | 8 million new users—transaction completion |
| 2014 | Polkomtel Sp. z o.o. | N/A | 100 | 0 | Operational and financial synergy, market consolidation, economics of scale, client base expansion, product portfolio diversification | 8 million new users—transaction completion |
| 2015 | Muzo fm Sp. z o.o. | 1.3 | 100 | 0 | Product portfolio diversification | 400 thousand daily listeners, radio frequency |
| 2015 | Redefine Sp. z o.o. | N/A | 100 | 0 | Market consolidation, sales channels expansion, product portfolio diversification | 1.4 million active users—transaction completion |
| 2016 | Litenite Ltd. | 197.3 | 100 | 0 | Operational and financial synergy, economics of scale, market consolidation | Acquisition and debt cancellation |
| 2016 | Midas S.A. | 81.74 | 27.24 | 65.99 | Operational and financial synergy, economics of scale, market consolidation | LTE frequency acquisition |

**Table 3.** *Cont.*

| Date of Final Agreement | Target | Transaction Value (mln USD) | Shares Bought (%) | Shares Before Transaction (%) | Business Reason | Client Base/Key Assets Captured |
|---|---|---|---|---|---|---|
| 2016 | Midas S.A. | 20.53 | 6.76 | 93.24 | Operational and financial synergy, economics of scale, market consolidation | LTE frequency acquisition—transaction completion |
| 2016 | Netshare Sp. z o.o. | N/A | 100 | 0 | Operational and financial synergy | LTE frequency acquisition—transaction completion |
| 2017 | Eileme 1,2,3,4 AB | N/A | 199 | 0 | Operational and financial synergy, sales channels expansion, product portfolio diversification | 1.4 million active users—transaction completion |
| 2018 | Netia S.A. | 195.04 | 33 | 0 | Client base expansion, product portfolio diversification, operational and financial synergy | 1 million client base acquisitions, over 800 of the biggest business clients in Poland |
| 2018 | Netia S.A. | N/A | 33 | 0 | Client base expansion, product portfolio diversification, market consolidation | 1 million client base acquisitions, over 800 of the biggest business clients in Poland—transaction completion |
| 2018 | Eleven Sports Network Sp. z o.o. | 44.58 | 50 | 0 | Product portfolio diversification, operational and financial synergy, client base expansion | 2 million active paying users and subscribers, licenses |
| 2019 | Vindix Sp. z o.o. | N/A | 0 | 0 | Operational and financial synergy | 77 debt portfolios, worth over PLN 90 million |
| 2019 | Asseco Poland S.A. | 312.05 | 21.95 | 1.05 | Operational and financial synergy | ICT software development company and know-how |
| 2020 | Alledo Sp. z o.o. | N/A | 0 | 0 | Product portfolio diversification, new resources acquisition, operational and financial synergy | Solar panels and LED light producer and know-how; cross/up-sell opportunities |
| 2020 | Grupa Interia PL Sp. z o.o. | N/A | 100 | 0 | Product portfolio diversification, operational and financial synergy, market consolidation | 16 million daily active users |
| 2020 | Bcast Inc | 1.66 | 69 | 0 | New resources acquisition, operational and financial synergy, economics of scale | ICT software development company and know-how |
| 2020 | Interia.pl | N/A | 100 | 0 | Operational and financial synergy, market consolidation, client base expansion, product portfolio diversification | 16 million daily active users—transaction completion |
| 2020 | Spektrum TV Kozep- Europai Musorkes zito zrt | N/A | 50.52 | 49.48 | New resources acquisition, operational and financial synergy, economics of scale, market consolidation | TV Channels and shows |

Source: own, based on press releases, company reports.

As seen above, Cyfrowy Polsat Group consecutively expanded its client base, ensuring the acquisition of core assets, enabling horizontal and vertical integration. For instance, the acquisition of the Ipla.tv on-demand streaming platform, previously a satellite-based TV service, moved towards digital to strengthen the market position of Cyfrowy Polsat as an aggregator and content distributor. Another example is the acquisition of Polkomtel or Netia—Telco and broadband providers, allowing Polsat to obtain synergy effects from convergent offerings to their B2C clients, i.e., TV, Internet and phone in one bundle. Those acquisitions significantly expanded its customer base. In 2020, Cyfrowy Polsat Group had around 17 m regular users or clients, of which 25–30% are so-called "multiplay" clients, using more than two products at once [49]. The monetization of acquired client base, supported with 4.0 technologies, enables Cyfrowy Polsat Group's growth, despite the relatively low dynamics of the media industry trajectory. Another example is the acquisition of Vindix, a debt collection company—instead of selling unprofitable clients to other companies, Cyfrowy Polsat Group can manage the liabilities of their clients more easily and efficiently. Until 2020, Cyfrowy Polsat Group had held an established position on Telco, TV and radio market; however, digital media content had not been penetrated. By acquiring Interia, Cyfrowy Polsat integrated vertically, and has every type of media

company in their portfolio. Cyfrowy Polsat Group also moved towards becoming the first integrated media devices company, as they also operate in the utilities and energy sector. Cyfrowy Polsat Group bought Alledo—light and photovoltaic energy producer—and ZE PAK—lignite power plant energy producer. As evidenced above, Cyfrowy Polsat Group's strategy has remained unchanged for the last 10 years. Through mergers and acquisitions, at the beginning it established a large group of customers in the Polish market. Then, it moved into portfolio diversification and offered its clients more and more complex products and services, deriving from the achieved network effect. This was also often communicated by management in the press releases, justifying the acquisitions.

The company board confirms this strategy, especially after the acquisition of Polkomtel (Plus GSM), when KPI reporting has changed. Instead of reporting basic average revenue per user numbers, as most subscription-based companies do, Cyfrowy Polsat Group started to focus on revenue-generating unit saturation, which meant how many active services one client has. Between Q1 2018 and Q3 2020 (last reported period), the RPU grew by 13% to 2.72 points, proving that the strategy of leveraging effects of scale, monetizing the client base, and diversifying product portfolio works [50].

Moreover, when announcing new acquisitions, the Cyfrowy Polsat board communicates the operational and financial synergy externalities, presenting a conscious understanding of its multi-level BM 4.0 strategy. During the acquisition of Netia, a broadband provider, Cyfrowy Polsat published: "the synergy effects of the acquisition for revenue, costs, CAPEX are estimated at 800 million of PLN between 2018 and 2023" [49].

Another example of network rent realization is the acquisition of Interia. From the user perspective, the ads for Polsat TV shows are displayed as pop-up ads on Interia, delivered over the Netia fiber-optic Internet; clients see shows on Polkomtel provided smartphones through Ipla or Cyfrowy Polsat satellite TV, while also receiving an opportunity to install solar panels on their roofs. Average revenue per user of the whole group grows by 5–7% yearly and is one of the key performance indicators reported by the group.

Among M&A, we can also find examples of buying new technology 4.0 businesses. These are i.a. acquisitions of LItenite LTD., Midas S.A., NetShare sp. z o.o., Vindix sp.z o.o., Asseco, Bcast Inc. According to the owners' declarations (reports for shareholders), these acquisitions had an impact on reducing transaction costs (minimizing), accelerating customer service processes, increasing the use of AI in reducing human work and increasing the speed of customization of services and building scalable customer service systems. It is also worth emphasizing that, in the strategies of M&A, Cyfrowy Polsat S.A. tries to take over businesses with mature business models (Ipla, Netia S.A., Polkomtel S.A.) and advanced technology. Among the group's recent acquisitions, there are no insolvent businesses or companies that do not add a certain value in the field of technology.

The analysis, explained in details in Table 4., was carried out independently for years 2015–2019 as this period was the only one fully available due to the change in IFRS 15—Revenue from Contracts with Customers reporting standards. In general, Cyfrowy Polsat manages to retain over 30% EBITDA margins throughout the years, despite their constant decline of 15%. Despite worsening sector profitability, revenue per client increased by 30% from PLN 194.78 to 253.49, while subscription client base remained relatively flat, increasing by only 1%. If we take under the consideration that Cyfrowy Polsat lies in the intersection of ICT/Media business and competes with other TV broadcasters and Telco operators, Cyfrowy Polsat stays strong in benchmark comparisons. The group's biggest private competitor in the TV sector is the TVN Group, which regularly publishes negative bottom-line reports, while Polsat maintains at least 10% profit throughout the years. When it comes to Telco operators, the market is consolidated between four players: T-Mobile, P4, Orange, and Polkomtel (Plus) belonging to Cyfrowy Polsat. According to the Office of Electronic Communications in Poland, the market shares changed due to strong pressure from T-Mobile, and Orange, but Polkomtel managed to retain most of its customers base, while P4 suffered the most. As seen in Table 5. Polkomtel network, belonging to Cyfrowy Polsat, still remains stable telecom market shares, despite strong market competition.

**Table 4.** Financial and KPI results of Cyfrowy Polsat between Q1 2015 and Q4 2019.

| Quarter and Year | Subscription Clients (mln) | Revenue (bn PLN) | Net Profit (mln PLN) | EBIDTA (mln PLN) | EBITDA Margin | Revenue per Client (PLN) |
|---|---|---|---|---|---|---|
| 1Q2015 | 11.957 | 2.329 | 170.9 | 896.6 | 38.50% | 194.78 |
| 2Q2015 | 11.933 | 2.469 | 305 | 977 | 39.60% | 206.91 |
| 3Q2015 | 11.939 | 2.415 | 502.5 | 930 | 38.50% | 202.28 |
| 4Q2015 | 11.934 | 2.61 | 186 | 891 | 33.80% | 218.70 |
| 1Q2016 | 11.936 | 2.364 | 179 | 846 | 35.80% | 198.06 |
| 2Q2016 | 11.971 | 2.443 | 231 | 935 | 38.30% | 204.08 |
| 3Q2016 | 11.822 | 2.11 | 270 | 957 | 40.10% | 178.48 |
| 4Q2016 | 11.679 | 2.535 | 342 | 902 | 35.60% | 217.06 |
| 1Q2017 | 11.382 | 2.389 | 271 | 929 | 38.90% | 209.89 |
| 2Q2017 | 11.381 | 2.47 | 282 | 964 | 39.00% | 217.03 |
| 3Q2017 | 11.464 | 2.391 | 235 | 851 | 35.60% | 208.57 |
| 4Q2017 | 11.498 | 2.579 | 157 | 873 | 33.90% | 224.30 |
| 1Q2018 | 11.52 | 2.346 | 292 | 890 | 37.90% | 203.65 |
| 2Q2018 | 11.611 | 2.603 | 231 | 946 | 36.10% | 224.18 |
| 3Q2018 | 11.723 | 2.735 | 227 | 920 | 33.60% | 233.30 |
| 4Q2018 | 11.712 | 3.002 | 66.1 | 941 | 31.40% | 256.32 |
| 1Q2019 | 11.752 | 2.792 | 300 | 1038 | 38.20% | 237.58 |
| 2Q2019 | 11.913 | 2.923 | 269 | 1076 | 36.80% | 245.36 |
| 3Q2019 | 12.09 | 2.892 | 236.5 | 1021 | 35.30% | 239.21 |
| 4Q2019 | 12.107 | 3.069 | 311.9 | 1062 | 32.70% | 253.49 |

Source: own elaboration based on company reports, only for fully available years with new IFRS 15 reporting standards.

**Table 5.** Share of mobile phone operators based on revenue generated in Poland in 2018 and 2020.

| Revenue | 2018 | 2020 | Change (p.p.) |
|---|---|---|---|
| T-Mobile | 21.10% | 24.30% | 3.20 p.p. |
| P4 | 26.10% | 20.20% | −5.90 p.p. |
| Polkomtel | 25.10% | 24.90% | −0.20 p.p. |
| Orange | 24.50% | 28.30% | 3.80 p.p. |

Accessed on 2 September 2021; Source: Office of Electronic Communication.

Cyfrowy Polsat as a group needs to utilize network-based strategy as the communications sector is already saturated. In 2020, the penetration rate of phones and internet connection consecutively reached 140% and 90% leaving limited opportunity for growth (Office of Electronic Communication). Similarly, the infrastructure costs, mostly due to data consumption and 5G investments, increased by 55%. The Polish IT Chamber projections suggest exponential 5–7× growth, creating stronger cost pressure on providers.

When it comes to prices, we observe a pricing pressure on Polish peers. For instance, a US or Western European citizen pays more for data charges, and rarely enjoys an unlimited data plan, while in Poland this is part of the regular packaging. Annual revenue per user decreased at −24% CAGR between 2015–2020; however, the bundled offerings began to be far more important, and they are the main revenue drivers for both Cyfrowy Polsat and Orange. In fact, these two players changed their reporting to emphasize the importance of bundle sales model, in which one customer receives a multiproduct offering: from fiber, through TV, ending with a telephone plan for a four-person family. As seen in the Figure 2 below, we observe over 5–7× revenue differences between one offering and convergent

ones. This allows for retaining profitability as customer acquisition cost remains similar in any case.

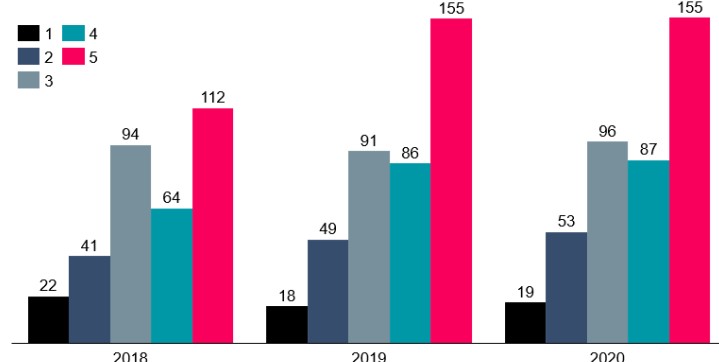

**Figure 2.** Average monthly revenue per user from bundled telecommunication services (ARPU) in Poland from 2018 to 2020. Source: Office of Electronic Communication.

This proves that network effects were leveraged to fully monetize the customer base, which otherwise would remain stagnant. By acquiring new companies, the Cyfrowy Polsat Group manages to increase its client base, overcoming organic growth obstacles, and realizing synergies of cross/up-sell opportunities within product portfolio.

In 2019, the Cyfrowy Polsat Group bought a minority stake in Asseco Poland. It was a signal to the market, that the Group was looking for solutions to strengthen its IT infrastructure. For the Cyfrowy Polsat, Asseco started to be the main technological partner supporting the development and maintenance of IT systems. The commercialization of 5G networks, artificial intelligence used for customer data analytics 4.0 industry, cyber security or big data are just a few examples of areas where both companies could achieve synergy. On top of that, Asseco is a leading software hub in Poland, being the key technological partner of Cyfrowy Polsat. Despite the purchase of only 21.95% of Asseco shares, it was the second largest investment of Cyfrowy Polsat (USD 312.05 million) which clearly emphasized the strategic importance of this investment. Investment in Asseco was important because of two aspects: creating a technological advantage, and to capture value. Previously, Asseco was key technological partner of the biggest Telco competitor, Orange Polska. According to Rafal Kozlowski, VP of Asseco: "The most important part is to increase level of customer service. The 5G aspects are the next technological challenges, which are currently analyze" [51].

Thanks to its intense M&A activity on the Polish market, the Cyfrowy Polsat Group acquired entities from various areas of activity: broadcasting and production of television, Internet media, telecommunications, pay TV, online video and integrated services. Although they coincide, the key factor for Cyfrowy Polsat Group was the effective and efficient integration of the acquired resources in order to search for the synergy effect that would guarantee the creation of added value, primarily for its shareholders. Cardiam et al. emphasized that resource integration is an embedded process of matching (fitting of available resources and primarily concerns interaction), resourcing (resource creation, integration and resistance removal) and valuing (assessment of value in the social context, the determination of positive or negative outcomes from the enactment of resourcing) [52]. Those concepts have been seen in Cyfrowy Polsat Group's successful strategy: precise assessment of the potential target of the acquisition (Metelem Holding), creation of the new content (Telewizja Polsat), its further distribution through the distribution channels (mPunkt Polska or Ipla), supported by constant resource development (Asseco Poland). The measurable effect of this success is reflected in its the share price.

Since its IPO in 2013, the Cyfrowy Polsat stock price grew from PLN 17 to PLN 29, maintaining healthy profitability, and stable growth rates, outpacing the WIG stock index by 72% within the last ten years, which was presented in the Figure 3. The Beta coefficient of Cyfrowy Polsat S.A. to WiG for the last 10 years is 0.843 according to the financial

data published in the Company's reports. A beta coefficient can measure the volatility of an individual stock compared to the systematic risk of the entire market. A beta value that is less than 1.0 means that the security is theoretically less volatile than the market. Including this stock in a portfolio makes it less risky than the same portfolio without the stock. The outperformance is especially visible from early 2013 when Cyfrowy Polsat started to regularly acquire other companies, realizing its programmatic M&A strategy. The group is also a main vehicle of wealth growth for main owner, Zygmunt Solorz-Żak, who is consecutively the richest Pole, in various rankings.

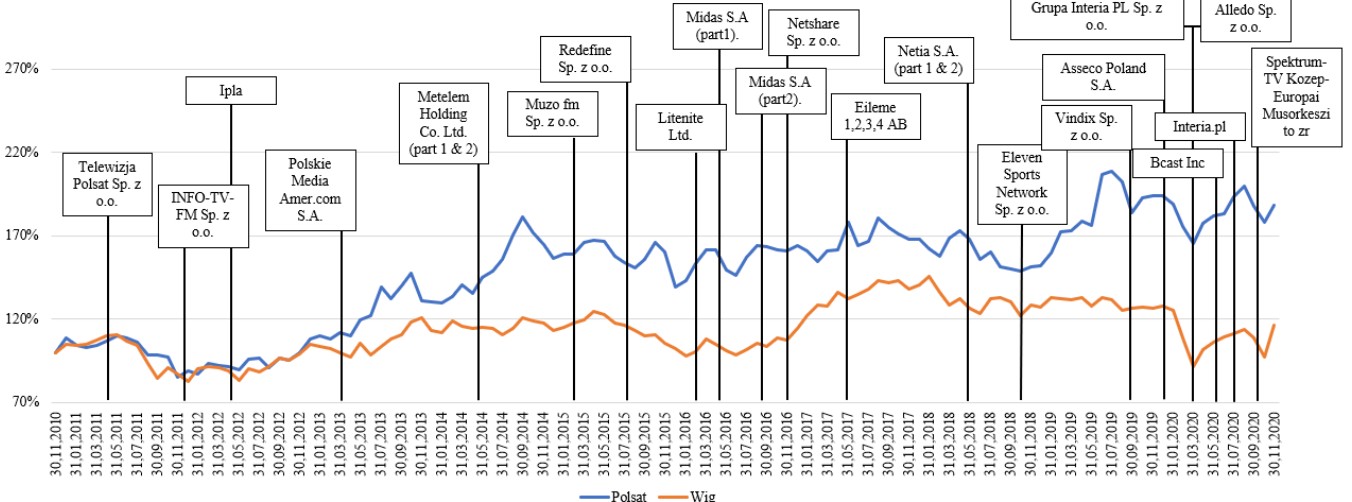

**Figure 3.** Cyfrowy Polsat stocks performance vs. main Warsaw Stock Index (WIG) vs. M&A activity of Cyfrowy Polsat S.A., between 30.11.2010 and 30.11.2020. The baseline is 100%; Source: own elaboration based on stooq.pl.

## 5. Conclusions and Discussion of the Results

The aim of the study, the results of which are presented in this article, was to indicate the possibilities of using a business model from the perspective of Industry 4.0 based on the theory of the network effect in building the value of network organizations.

The main objective of the study, the results of which are presented in the article, was to indicate the possibility of using a business model from the perspective of Industry 4.0 based on the theory of the network effect in building the value of network organizations. It is worth emphasizing that the network effect was first used by Jeffrey Rohfls. He investigated the telecommunication sector. The utility that a subscriber derives from a communications service increases as others join the system. He conducted his research in the 1970s. A lot has changed since then. Mainly due to the development of technology, the network effect can also be found in other sectors. However, a completely new impetus was given by the development of Industry 4.0, thanks to which possibilities of creating network effects apply to most sectors [53]. The conducted research allows one to conclude that, especially 4.0 technologies, such as AI, Cloud Computing, big data and IoT have an impact on the shaping of a new business model in the telecommunication sector (the first research question). The new model offers qualitatively different possibilities of achieving the network effect without incurring excessive transaction costs and with new possibilities of building economies of scale. The impact on the reduction, elimination of transaction costs and new possibilities of model scalability are the main features of the method of increasing the possibility of obtaining network effects (second research question). Thanks to business models 4.0, organizations in the telecommunication sector can (as was shown in the case study) move from a classic subscription business model to a model based on the network effect [53]. In the future, the development of this model will be a business model containing the features of the business ecosystem [54] with elements of complexity theory [55]. G. Parker, who in his research proves that "platform ecosystems rely on

economies of scale, data-driven economies of scope, high quality algorithmic systems, and strong network effects that typically promote winner-take-most markets" [56].

The presented case of a media group with its M&A confirms that 4.0 technologies implemented in the business model can change not only quantitatively (the number of customers), but also qualitatively (new monetization schemes—the way of operation).

The results of the study of a class 4.0 business model using network effects presented in the article meet the requirements for the study. It presents the way technology 4.0 influences building the value characteristic of business models and the impact of these categories on the possibility of using direct and indirect network effects. The case report of the Cyfrowy Polsat Group contained in the article helps to facilitate an understanding of the mechanisms of this impact by identifying the key direct effects of the relationship between building value and the increase in the number of customers, the value measured by the size of the markets served. The results of the study carried out in accordance with the case study method can be generalized only to cases with similar properties. This is a fundamental limitation of the case study. Nevertheless, the analyzes and prepared conclusions allow us to confirm the usefulness of researching business models that refer to the concept of the network effect and are technologically supported by innovations from Industry 4.0. The research used the analysis of the IFRS index is characteristic of organizations providing services to clients and groups of clients and measuring the effectiveness of management in this type of activities. Of course, the authors know the limitations of this indicator. Nevertheless, in the situation of the network effect analysis, it is important to include the entire effect of interactions, and the continuous analysis is able to strengthen the credibility of the result obtained. To some extent, the conclusion of the authors of the study is confirmed by comparing the company's results with the general stock market trend. In subsequent studies, it is worth subjecting the rigors of the research method to a wider representation of companies, also in other sectors, and in this way to comprehensively check the usefulness of 4.0 business models based on the concept of the network effect.

The presented case study also allows for the indication of the formation of indirect effects resulting from the use of SaaS services, sharing economy platforms, and ride sharing platforms. Acquisitions of companies increasing the possibilities of servicing IT infrastructure is an opportunity to ensure in the long term an increase in the number of serviced customers at a lower transaction cost [57]. The indicated example can also be treated as a form of organizational synergy analysis [58]. It is also an example of focusing on building an organization focused on meeting the needs of a diverse group of stakeholders [59].

Rebound changes in the Cyfrowy Polsat group "we want to be the leader of the entertainment and telecommunications market in Poland, we want to do so using the best and most modern technologies in order to provide high-quality integrated services, and the overriding goal of our strategy is sustainable, long-term value growth" [60] indicate a consistent strategy of building value based on an increase in the number of customers and an increase in Revenue from Contracts with Customers.

In the communication services sector, the network effect has been the main motive of integration for many years. However, it was not called a network effect, and was often referred to as a strategy of building new markets [61], a strategy of horizontal diversification, strategies of resource diversification [62,63]. This was mainly the case of earlier research. Analyzing very broadly, it was mainly a problem of some kind of diversification. In this way, the building of this type of organization in industry analysis and resource-based management was analyzed. In these approaches to the strategy, the main emphasis was on building the organization's revenues by increasing market shares or increasing the value of the organization as a result of M&A itself. Only the network school in strategic management pointed to other possibilities of building the value of the organization, emphasizing the relational resources of various groups of stakeholders and their impact on building specific feedbacks in the system of supporting income streams.

The next stage, symbolically marked in this article, will be an attempt to interpret network effects as the basis for shaping the ecosystems of an organization [64].

The theory of business models and the development of 4.0 technologies providing the basis for the definition of new 4.0 business models (as presented in the case of Cyfrowy Polsat in the Figure 4), in the opinion of the authors of the article, confirmed by the literature research and case study analysis, will probably be the main factor accelerating the transition to the interpretation of network as the basis for shaping management ecosystems in the spirit of complexity theory.

Changes in the perception of the network effect in management evolve similarly to the already historic outsourcing. Both of these categories first appeared in informatic sciences and only later were adopted by management sciences. Similarly, it has also evolved in the management as the concept of economic platforms [65].

It also emphasized the importance of the value creation from the resource integration in and resource interaction across the Cyfrowy Polsat Group's. The efficient integration of the resources acquired through many of its M&As was crucial to meet shareholder's high expectation. Its strategy clearly represents some of the key concepts of the Resource Interaction Approach, described by Baraldi et al. as channelling the newly acquired services through the same interface, resource combinations across the Group's entities, innovations and technical developments [66].

**Meta network of Network Effects for Cyfrowy Polsat**

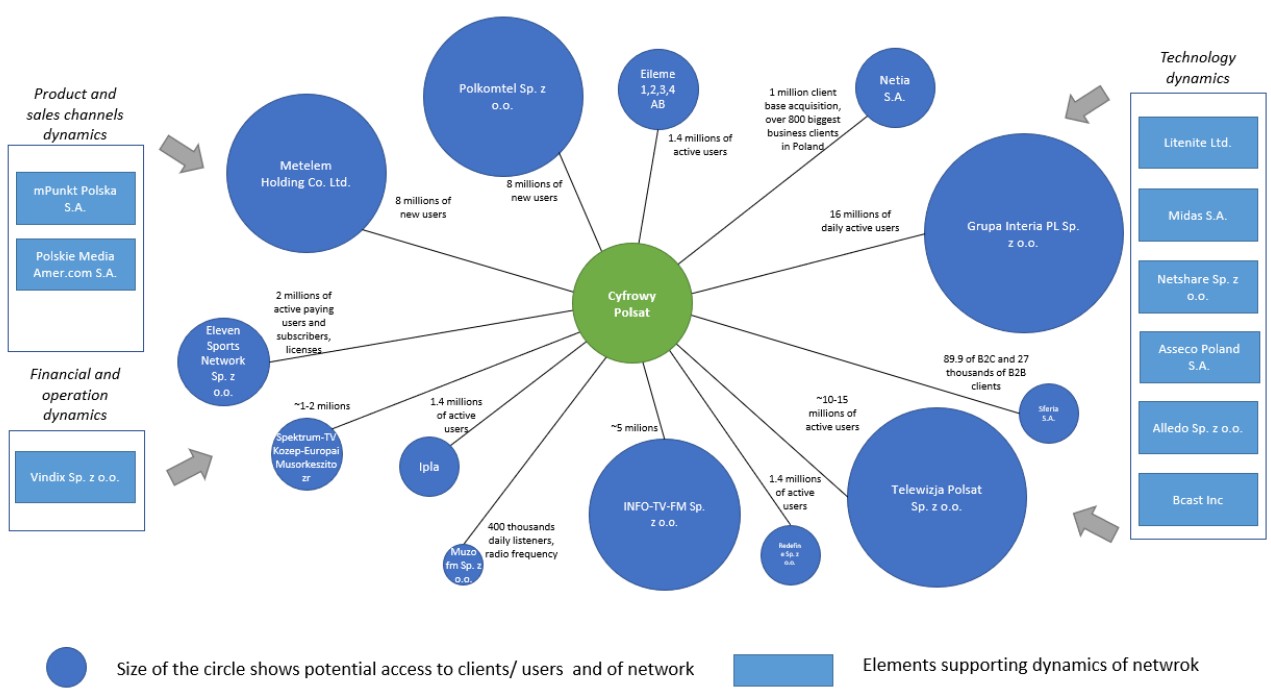

**Figure 4.** Meta network of network effects for Cyfrowy Polsat; source: own.

It is worth mentioning also that further acquisitions and building network with users and product and services will have impact on network effect. Network effect, in this case, is measured by IFRS 15—Revenue from Contracts with Customers. This metric, according to the authors, may be a good indicator to show the potential advancement of the network effect in a given organization. In this case, if the Contracts with Customers indicator is growing in the proper period, that can mean that the network effect is stronger, and it is more intense. This indicator is most often used in analyses of organizations that build value by adding new customer groups (M&A) and by adding new services or products for existing customer groups (also through M&A).

Of course, the increase in value in the organization may be the result of many factors, dependent and independent of the company. However, the purpose of the study presented

in this article was for those factors that were associated with the expansion of the customer base and the increase in Revenue from Contracts with Customers. The results obtained for the studied group should be treated as reliable, not only because the data were obtained from public documents of the organization and processed in accordance with the logic of this indicator, but also because they are consistent with the strategic goals and strategic motives of the by them M&A.

The presented model does not exhaust all the possibilities of the impact of technology 4.0 on the nature of the business model, nor does it exhaust all possible variants of network business models. However, it is noteworthy that class 4.0 solutions allow for building new and more efficient business models, and often also lead to a change in the general concept of running a business.

As presented within the case study, by leveraging 4.0 business models, including SaaS, sharing economy, and IT infrastructure, alongside with increased volume of clients served, the Cyfrowy Polsat Group realizes a network rent. The company does it mostly through programmatic M&As of selected ICT/media companies, which complement their business model. With that, Cyfrowy Polsat Group monetizes its otherwise-stagnant subscription customers base by adding more connections of products towards customers. Moreover, it does this despite worsening conditions, constantly outperforming the market. It uses a similar integration strategy as a top technology company in the world, including Alphabet (Google), Amazon or Facebook. In this sense, it is an outstanding example of modern BM 4.0 within Central Eastern Europe.

The obtained results can be the basis for changes in the management of organizations. If indeed the transaction costs in the business model 4.0 using network effects can be reduced to zero, then hierarchical organizations so far dominant in management will be ineffective and will lose competition with organizations built on contracts. Then, it will also be possible to fully use the theory of complexity to define the organization's strategy [56].

Until then, managers in organizations can effectively use the concept of including new customer groups and new products for these customers in the organization, counting on the growth of Revenue from Contracts with Customers. The theory of network effects used in the strategies of Google, Facebook, the Netflix group and others has so far proved its advantage over other classic strategies from the concepts of I. Ansoff or M. Porter. However, it differs from the Google, Facebook and Netflix strategy in that it is implemented in a company that deals with media and telecommunications, but also operates in the energy, banking, insurance and real estate sectors. All these sectors operate using the described business model, showing that network effects can also be used cross-sectorally. The closest to this is Alphabet's development. Many other organizations, especially in the service sector, are already consciously moving in this direction.

**Author Contributions:** The authors contributed equally to this work. J.N., R.T., M.W., K.B. contributed to the conceptualization, formal analysis, investigation, methodology, writing of the original draft, and review and editing. All authors have read and agreed to the published version of the manuscript.

**Funding:** This research received no external funding.

**Institutional Review Board Statement:** Not applicable.

**Informed Consent Statement:** Not applicable.

**Data Availability Statement:** Publicly available datasets were analyzed in this study. This data can be found here: www.stooq.pl/q/d/?s=cps&c=0&d1=20101130&d2=20201130 (accessed on 20 September 2021) and here: www.stooq.pl/q/d/?s=wig&c=0&d1=20101130&d2=20201130 (accessed on 20 September 2021).

**Conflicts of Interest:** The authors declare no conflict of interest.

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
