# Peer review of "Business Models 4.0 Using Network Effects: Case Study of the Cyfrowy Polsat Group"

_sustainability, doi:10.3390/su132111570_

Round 1

Reviewer 1 Report

Some details like:

  • figure 1 - column 3. I guess the head shouldn't be "value proposition"? Rather "value management"?
  • table 1 nad 2 - it's totally unreadable
  • table 4 - I guess the name of comapny in last row is cut

My major concern is research method. I am not sure if M&A made by Cyfrowy Polsat can be viewed as introduction of BM 4.0. Of course, there are some sings of effects mentioned in article.

But M&A have been a general strategy of growth for companies from different industries for decades. Can we assign Network effects mainly to BM 4.0?

Moreover - can we define effectiveness of BM 4.0 only by value increase? There are so many factors influencing market value of company. Their influence hasn't been pointed out in paper.

I would try to examine such indicators as innovations, level of automatization of business processes, scale of AI use in Cyfrowy Polsat etc. In my opinion, that would give a broader view of BM 4.0 effectiveness.

Final comment - in my opinions there is lack of previous empirical research review.

Reviewer 2 Report

The paper presents interesting ideas for the growth of knowledge on the subject under study. However, in order to be published it needs major revisions following the indications below: - In the introduction should be included specific research questions; - In the paragraph of the methodology of research are missing citations related to the method of case studies; - The review of the literature should be enriched because it is scarce and outdated; - In the paragraph of the conclusions should be indicated in more detail the managerial implications; - All the work, starting from the abstract should be reread to eliminate redundancies and improve the quality of English.

Reviewer 3 Report

Using a real life case study, this paper intends to explain business models 4.0 using network effects. This is indeed an interesting topic, but quality of the paper is not satisfactory. I have a number of concerns that result in rejecting this paper.

Authors personal opinion is not good enough reason for conducting this research. The whole introduction is unconvincing and does not provide arguments to read the paper further. The aim is confusing – is it realistic that you can discover/expand new knowledge with a single case study?

Table 2 seems like repeating the Table 1. Further, what does Table 3. intend to show? Table 3 does not present change of business models, it just present the current business models of selected companies. Did you want to present the evolution of business models? Still, there are different sectors to compare.

Methodology (i.e. research procedure) is a problem too. I do not see what is really done in this study? What is “classic case study method”? Any reference? what type of analysis did you apply? Where are the results of correlation you mention? What do Scopus, Science Direct, JSTOR and Google Scholar databases have with your empirical study?

Results are mainly descriptive. Can you on the basis of one case study prove anything? The fact that “despite worsening sector profitability, revenue per client increased by 30% from 194,78 PLN to 253,49, while subscription client base remained relatively flat, increasing by only 1%” does not provide a proof that “network effects were leveraged to fully monetize the customer base, which otherwise would remain stagnant.” This can be a result of other factors too. What about the prices, operating costs or number of employees and their salaries in the analyzed period? Any change/increase? This section reads like a professional paper.

Major concern relates to the discussion and conclusion. Discussion is weak, not all business model elements are discussed, and interpretation of results in the light of the previous literature is missing. Research limitations, theoretical, policy and managerial implications are not mentioned.

The list of references is too short. More work can be done to strengthen the theoretical background and discussion.

Therefore, I suggest rejection of this paper.

Round 2

Reviewer 2 Report

Now is acceptable in present form

Author Response

Dear Reviewer,

Thank you very much for all comments.

Reviewer 3 Report

From my point of view, authors made little progress and it is still not enough to be published. All I can see is that list of references is expanded. New text is added without criticism. Starting from the abstract, the paper’s aim/purpose is not clear. You are saying (in your reply) that in cases where there are no other cases available for replication, the researcher can adopt the single-case design. Yes, it is true, but is it the case with your paper? There are similar companies around the world. You mentioned Google, Facebook and the Netflix group as using the same strategy as Cyfrowy Polsat. I still do no see the results of the “study of the correlation” you mention in research procedure?  Discussion is expanded but mainly with describing, and there is no interpretation of results in the light of the previous literature. This makes this paper reads like a professional paper even more. Implications for academics and policy makers are weak.

Author Response

We introduced several references to the literature as a part of the discussion enrichment. In this way we validated the quality of our research’s results. In the case of "correlation", we made corrective changes in the text. The theory of network effects used in the strategies of Google, Facebook, the Netflix group and others has so far proved its advantage over other classic strategies from the concepts of I. Ansoff or M. Porter. However, it differs from the Google, Facebook and Netflix strategy in that it is implemented in a company that deals with the media and telecommunications, but also operates in the energy, banking, insurance and real estate sectors. All these sectors operate using the described business model, showing that network effects can also be used cross-sectorally.